# Streptococcal Arginine Deiminase Inhibits T Lymphocyte Differentiation In Vitro

**DOI:** 10.3390/microorganisms11102585

**Published:** 2023-10-19

**Authors:** Eleonora A. Starikova, Jennet T. Mammedova, Arina Ozhiganova, Tatiana A. Leveshko, Aleksandra M. Lebedeva, Alexey V. Sokolov, Dmitry V. Isakov, Alena B. Karaseva, Larissa A. Burova, Igor V. Kudryavtsev

**Affiliations:** 1Laboratory of Cellular Immunology, Department of Immunology, Institute of Experimental Medicine, 197022 St. Petersburg, Russia; 2Medical Faculty, First Saint Petersburg State I. Pavlov Medical University, 197022 St. Petersburg, Russia; 3Laboratory of General Immunology, Department of Immunology, Institute of Experimental Medicine, 197022 St. Petersburg, Russia; 4Laboratory of Biochemical Genetics, Department of Molecular Genetics, Institute of Experimental Medicine, 197022 St. Petersburg, Russia; biochemsokolov@gmail.com; 5Laboratory of Molecular Genetics of Pathogenic Microorganisms, Department of Molecular Microbiology, Institute of Experimental Medicine, 197022 St. Petersburg, Russia; 6Laboratory of Biomedical Microecology, Department of Molecular Microbiology, Institute of Experimental Medicine, 197022 St. Petersburg, Russia; lburova@yandex.ru

**Keywords:** memory T cells, proliferation, differentiation, autophagy, arginine, *Streptococcus pyogenes*, arginine deiminase

## Abstract

Pathogenic microbes use arginine-metabolizing enzymes as an immune evasion strategy. In this study, the impact of streptococcal arginine deiminase (ADI) on the human peripheral blood T lymphocytes function in vitro was studied. The comparison of the effects of parental strain (*Streptococcus pyogenes* M49-16) with wild type of *ArcA* gene and its isogenic mutant with inactivated *ArcA* gene (*Streptococcus pyogenes* M49-16del*ArcA*) was carried out. It was found that ADI in parental strain SDSC composition resulted in a fivefold decrease in the arginine concentration in human peripheral blood mononuclear cell (PBMC) supernatants. Only parental strain SDSCs suppressed anti-CD2/CD3/CD28-bead-stimulated mitochondrial dehydrogenase activity and caused a twofold decrease in IL-2 production in PBMC. Flow cytometry analysis revealed that ADI decreased the percentage of CM (central memory) and increased the proportion of TEMRA (terminally differentiated effector memory) of CD4+ and CD8+ T cells subsets. Enzyme activity inhibited the proliferation of all CD8+ T cell subsets as well as CM, EM (effector memory), and TEMRA CD4+ T cells. One of the prominent ADI effects was the inhibition of autophagy processes in CD8+ CM and EM as well as CD4+ CM, EM, and TEMRA T cell subsets. The data obtained confirm arginine’s crucial role in controlling immune reactions and suggest that streptococcal ADI may downregulate adaptive immunity and immunological memory.

## 1. Introduction

Arginine is the most recognizable example of amino acid, whose catabolism is an important mechanism of fine-tune-up of immune reactions in mammals [1]. Arginine bioavailability, synthesis, and catabolism may result in diverse pro- or anti-inflammatory outcomes [2,3]. Studies with arginase-expressing myeloid cells [4,5], soluble arginase [6], or arginine-depleted medium demonstrate that arginine bioavailability plays a crucial role for efficient T cell activation [7]. In particular, the amino acid deficiency suppresses TCR downstream signal transduction, which provides T cell proliferation and differentiation [8,9,10]. Arginine availability modulates CD3ξ subunit expression coupling TCR to the downstream signaling cascade [3]. It was uncovered that stimulation of T cells in arginine-free settings results in downregulation of cyclin D3 and cyclin-dependent kinase 4 (cdk4) and reduces Rb protein phosphorylation and E2F1 expression, as well as arrests proliferation at the G0–G1 phase of the cell cycle [3].

Arginine deiminase (ADI) is a bacterial enzyme hydrolyzing arginine to citrulline and ammonia [11]. It was previously established that the enzyme can be expressed on the membrane of bacterial cells [12,13]. In addition, with increased consumption, the deficiency of intracellular arginine is replenished with ArcD arginine/ornithine antiporter [14,15,16]. Thus, ADI can, directly or indirectly through ArcD, exhaust the amino acid in the host cell microenvironment. In support of this, in a model of subcutaneous streptococcal infection in mice, a link was established between the arginine deiminase expression and a decrease in the plasma arginine concentration in infected animals [17].

Streptococci can use ADI to locally reduce arginine concentration in the most frequent persistence loci—the tonsils [18,19], which are peripheral organs of the immune system where clonal expansion and differentiation of T cells take place. It has been shown that as T cells are activated, their need for arginine increases dramatically [20]. Theoretically, even such a local arginine decrease can be sufficient to suppress immune cell functions. Pathogenic microbes expressing ADI may create metabolic background, resulting in alteration of immune reaction, nonproductive inflammation, and elevated risk of reinfection [21].

Bacterial ADI uses arginine as a substrate like host arginase does. However, compared to arginase, which converts arginine to ornithine, bacterial ADI catalyzes a conversion of arginine into citrulline [22]. Moreover, these enzymes exert distinct biochemical characteristics such as processivity, relative enzyme activity, and substrate affinity, as well as optimal pH value [23,24,25]. Therefore, molecular mechanisms underlying the activity of those enzymes on T cells may differ markedly. Experiments with arginine-free medium also vary from those with the use of arginine-hydrolyzing enzymes because the enzymes do not deplete a substrate entirely but generate respective metabolites. Based on this, investigating effects of bacterial ADI on T cell functions is a standalone relevant scientific issue. However, most studies assessing ADI effects are focused on examining the mechanisms of its antitumor activity [22,26]. Tumor cells exposed to ADI were shown to inhibit mechanical target of rapamycin (mTOR) pathway [27], aerobic glycolysis [28], and protein production [29], and enhance autophagy [30,31].

T cells circulate in vivo in a resting state, but upon antigen recognition and obtaining costimulatory signals they become activated to proliferate and acquire effector properties aimed at pathogen control and elimination. After pathogen elimination, the majority of effector T cells undergo apoptosis, whereas the remaining few form a pool of long-lived memory T cells [32,33,34]. A potential of T cells to maintain the memory of previous pathogen encounters allows the development of a more rapid, robust host immune response upon repeated contamination, thereby lowering the magnitude of tissue damage and the likelihood of repeated disease, per se [34]. Effector memory (EM) and terminally differentiated effector memory (TEMRA) T cells are mainly located in peripheral tissues, at sites of the most probable repeated contact with a pathogen [35]. Therefore, in case of reinfection, it is this T cell subset that should most probably end up in microenvironments featuring a nutrient deficit.

The nutrient-dependent mTOR signaling pathway and autophagy control key events in T cell activation, clonal expansion, and differentiation [36,37].

A close relation between T cell function and arginine metabolism suggest that bacterial arginine-hydrolyzing enzymes may serve as part of a pathogen immune evasion strategy. Upon that, studies looking into ADI impact on T cells remain relevant [38,39,40,41].

Here, we investigated the influence of streptococcal ADI on CD4+ and CD8+ T cell subset activation and differentiation. To study the bacterial enzyme, the earlier applied strategy was used [17,39,41,42]. The comparison between the effects of ultrasonic lysates of the parental strain expressing ADI with the effects of its isogenic mutant with an inactivated ADI gene was made. The relation of the enzyme effect to its ability to deplete arginine was verified by supraphysiological concentration arginine supplement. It was found out that ADI lowers arginine levels, resulting in inhibited autophagy, T cell proliferation, and decreased proportion of central memory (CM), along with elevated TEMRA CD8+ and CD4+ T cell subsets. Arginine supplement leveled the effect of the enzyme in most cases. Our data confirmed the pivotal role of arginine in developing adaptive immunity and revealed that bacterial ADI may affect memory T cell generation during infection.

## 2. Materials and Methods

### 2.1. Supernatants of Destroyed Streptococcal Cells

*S. pyogenes* M49-16 (paternal strain) was kindly provided by Dr J. Ferretti (Department of Microbiology, University of Oklahoma Health Sciences Center, Oklahoma City, USA). *S. pyogenes* M49-16del*ArcA* with inactivated *ArcA* gene was constructed as described previously [17]. Bacterial strains were kindly provided by Professor Suvorov A.N., Head, Department of Molecular Microbiology, Institute of Experimental Medicine. Parental- and mutant-strain-derived supernatant of destroyed streptococcal cells (SDSCs) were prepared by sonication from paternal and mutant strains according to the approach published earlier [17]. Previously, it was shown that significant ADI activity was present in parental SDSCs, but none could be detected in the mutant one [17].

### 2.2. Isolation and Culture of Human Peripheral Blood Mononuclear Cells (PBMCs)

The peripheral blood samples from volunteers aged 20 to 50 years were collected into tubes and added to anticoagulant K_3_EDTA. The human study was reviewed and approved by the Ethics Committee of the Institute of Experimental Medicine, permission No. 2/19, dated 25 March 2019. The PBMC isolation was carried out by using sedimentation in Ficoll gradient (Biolot, Moscow, Russia), density 1.077 g/mL. Blood samples were prediluted with Hanks’ solution (Biolot) 3 times followed by layering carefully onto Ficoll solution. Cells were separated in a centrifuge (Eppendorf, Hamburg, Germany) with a horizontal rotor for 40 min at 400× *g* and 21 °C. PBMCs were transferred from the interphase ring formed at Ficoll border, and washed twice in Versen’s solution (Biolot, Moscow, Russia) by centrifugation at 300× *g* for 15 min at 4 °C. Cell viability (at least 98%) was assessed by staining with a 0.2% trypan blue solution followed by counting live and dead cells in a hemocytometer. PBMC purification quality was checked with flow cytometry prior to the experiments. The leukocyte subset frequency is presented in Appendix A. Cells were resuspended and cultured in RPMI-1640 medium (PanEco, Moscow, Russia) supplemented with 10% inactivated fetal bovine serum (Invitrogen, Waltham, MA, USA), 2 mM glutamine (Biolot, Moscow, Russia), 50 μg/mL gentamicin (Biolot, Moscow, Russia), and 50 μM β-mercaptoethanol (Sigma-Aldrich, Waltham, MA, USA, ). Cellularity was adjusted to a concentration of 1 million/1 mL. Cells were cultured at 37 °C in a humid atmosphere with 5% CO_2_. Subsequently, the cell suspension was used to assess SDSC-related effects on cell activation, differentiation, and intensity of autophagy processes.

### 2.3. PBMC Activation Assessed in MTT Assay

To evaluate the effect of streptococcal ADI on PBMC activation, we assessed parental- and mutant-strain-derived SDSC-related effects on mitochondrial dehydrogenase activity. For that purpose, 100 μL of cell suspension (1 million/mL) was added to a 96-well flat-bottom plate (Sarstedt, Nümbrecht, Germany). The T Cell Activation/Expansion Kit (Miltenyi Biotec Inc., GmbH, Bergisch Gladbach, Germany), containing antibiotin antibody-coated beads and biotinylated antibodies against CD2, CD3, CD28 T cell surface molecules, was used for cell activation. Thus, T cell binding to antibody-loaded beads mimics its interaction with antigen-presenting cells. Original or mutant-strain-derived SDSCs were added at a dilution of 1/200 (*v*/*v*). L-arginine (Sigma) was added at a supraphysiological concentration of 2 mM to check whether the effect of arginine deiminase was associated with arginine depletion. Cell activation was assessed 24, 48, 72, and 96 h after the onset of experiment. For the last 4 h, cells were incubated with 0.5 mg/mL MTT (3-(4.5-dimethylthiazol-2-yl)-2.5-diphenyl-tetrazolium bromide) (Sigma) solution. For cell lysis and formazan crystal dissolution, 100 μL of lysis buffer containing 10% sodium dodecyl sulfate (Serva, Heidelberg, Germany) in 0.01 N-HCl was added into the wells and incubated for 18 h at 37 °C. Optical density was measured on a spectrophotometer (Bio-Rad, Tokyo, Japan), at a wavelength of 570 nm. The data were expressed as percentage, and the optical density of the control wells was considered as 100%.

### 2.4. Assessing SDSC-Mediated Effects on PBMCs IL-2 Production

After 96 h of incubation, PBMC suspensions were collected into microtubes (Eppendorf) and centrifuged for 5 min at 300× *g*. The culture medium samples were collected and stored at −20 °C. The assessment of IL-2 concentration in the samples was performed using a human IL-2 Enzyme-Linked Immunosorbent Assay (ELISA) kit (Cytokine, Moscow, Russia), following the manufacturer’s instructions.

### 2.5. PBMC Supernatant Arginine Concentration Measurement

The change in arginine concentration was assessed by a method based on the Sakaguchi reaction with modifications: arginine is able to form a red-colored compound while interacting with 8-hydroxyquinoline and sodium hypobromite in an alkaline medium. Cells were seeded in 96-well plates (Sarstedt, Nümbrecht, Germany) at a concentration of 100,000 in 100 μL of complete culture medium and incubated with SDSCs of parental and mutant strains. To check whether the action of arginine deiminase is associated with the depletion of arginine, an amino acid supplement was added at a supraphysiological concentration of 2 mM. After 24, 48, 72, and 96 h, cell suspensions were collected in microtubes (Eppendorf, Hamburg, Germany) and centrifuged for 5 min at 300× *g*. The supernatants were transferred into clean microtubes (Eppendorf, Hamburg, Germany), and the protein was precipitated by centrifugation at 13,000× *g* for 5 min in a 10% aqueous solution of ZnSO_4_ and 0.5 M NaOH. To assess arginine concentration, 100 µL of the supernatant was transferred into 96-well plates (Sarstedt, Nümbrecht, Germany), and 50 µL of 5 mM 8-hydroxyquinoline and 100 µL of 8 mM sodium hypobromite in 2 M NaOH solution were added to the samples. A 6% BSA solution was used as a negative control (arginine-free sample). Optical density was recorded using a plate spectrophotometer ClarioStar (BMG Labtech, Ortenberg, Germany) at a wavelength of 495 nm. For each experiment, a calibration plot of absorbance at 495 nm versus arginine hydrochloride concentration (range 4–500 μM) in PBS was obtained. Arginine concentration in the samples was calculated using the Mars software 3.01 R2 package supplemented with ClarioStar reader, taking into account the volume fraction of the culture medium in the analyzed sample.

### 2.6. Assessing SDSC-Mediated Effect on T Cell Differentiation

To assess the SDSC-mediated effect on T cell activation and differentiation, 500 μL of PMBC suspension (1 million/mL) prepared ex tempore was placed into a 24-well plate (Sarstedt, Nümbrecht, Germany). Induced cell differentiation was performed by using a T Cell Activation/Expansion Kit (Miltenyi Biotec Inc., Bergisch Gladbach, GmbH, Germany), according to the manufacturer’s recommendations. Next, the test substances were added into the wells according to the following scheme: SDSC—at concentration of 1/200 (*v*/*v*), arginine (Sigma-Aldrich, St Louis, MO, USA)—2 mM. Cells were incubated for 48 or 96 h at 37 °C in a humid atmosphere with 5% CO_2_. In the latter case, culture medium was partially replaced on day 3 by using all test substances. At the end of incubation, the cells were collected into tubes for flow cytometry (Sarstedt). To analyze percentage of cells at different differentiation stages of helper T (CD3+CD4+) and cytotoxic T (CD3+CD4−) cells, cell suspensions were stained with an antibodies cocktail: CD4 labeled with APC (Cat. No. IM2468); CD45RA labeled with PC7 (cat. no. B10821); CD62L labeled with ECD (cat. no. IM2713U); CD3 labeled with APC-Alexa 750 (cat. no. A66329) (all purchased from Beckman Coulter, Brea, CA, USA), according to the manufacturer’s recommendations. To exclude necrotic cells from the analysis, the samples were stained with DNA-binding dye DAPI (Invitrogen) at a concentration of 300 nM. Samples were analyzed on a Navios flow cytometer (Beckman Coulter). DAPI-negative cells were used for analysis. Naïve cells were defined as CD45RA+CD62L+, central memory cells (CM) as CD45RA−CD62L+, effector memory cells (EM) as CD45RA-CD62L−, and terminally differentiated effector memory cells (TEMRA) as CD45RA+CD62L−. The T cell subset ratio was expressed as a percentage.

### 2.7. Assessing T Cell Subset Proliferative Activity

Cell proliferation was assessed using a method based on intracellular protein staining with in vivo fluorescent dye carboxyfluorescein succinimidyl ester (CFSE) (Sigma-Aldrich). PBMCs isolated as described above were adjusted to a concentration of 1 × 10^6^ cells/saline mL (Biolot) containing CFSE (Sigma-Aldrich, St. Louis, MO, USA) at a final concentration of 0.5 μg/mL to be incubated for 10 min in a water bath at 37 °C. After that, the cells were washed twice to remove the dye by centrifugation at 300× *g*, 4 °C, for 15 min in cold Hank’s solution containing 1% FBS (Sigma-Aldrich, St. Louis, MO, USA). CFSE-stained cells were resuspended in complete culture medium RPMI 1640 (Biolot) containing 10% FBS (Invitrogen, South America), 50 μg/mL gentamicin (Biolot), 2 mM glutamine (Biolot), and 50 μM β-mercaptoethanol (Sigma-Aldrich, St Louis, MO, USA). Cells (2 million/mL) were added to the wells of a 24-well plate. To induce proliferation, a T Cell Activation/Expansion Kit (Miltenyi Biotec Inc., Bergisch Gladbach, GmbH, Germany) was used according to the manufacturer’s recommendations. Parental- and mutant-strain-derived SDSCs were added at 1/200 dilution, and arginine at a concentration of 2 mM (Sigma-Aldrich, St. Louis, MO, USA) was added to some wells. The cells were incubated for 96 h; on day 3, the culture medium was partially replaced with fresh culture medium containing test substances. After incubation, cells were collected into cytometric tubes (Beckman Coulter, Brea, CA, USA) and stained with antibodies as described above. FCS Express 7 (De Novo Software, Pasadena, CA, USA) demo software was used to analyze flow cytometry data, expressed as division index (DI), which reflects the average number of cell divisions undergone by cells in the original population, including the fluorescence peak of nonproliferating cells.

### 2.8. Assessing ADI Impact on T Cell Autophagy

To assess autophagy in PBMCs stained with antibody cocktail (see Section 2.6), samples were added with dye Lysotracker Green DND-26 (Invitrogen, Eugene, OR, USA) at a concentration of 50 nM for 15 min. Cells were washed out by centrifugation in 1 mL PBS, for 5 min, at 300× *g*. To exclude necrotic cells from the analysis, the samples were stained with DNA-binding dye DAPI (Invitrogen, Rockford, IL, USA) at a concentration of 300 nM. Autophagy level was expressed as MFI. Anti-CD2/CD3/CD28-loaded bead treatment for 48 h was used as positive control, and autophagy inhibitor chloroquine (10 μM) was applied for the last 20 h of incubation as negative control (Appendix A).

### 2.9. Statistical Analysis

Statistical analysis and graphic presentation were performed using the software programs Prism 8.0 (GraphPad Software, Inc., San Diego, CA, USA), Microsoft Excel 2010, Statistica 7.0, Kaluza Analysis 2.1 and Navios Software 2.0 (Beckman Coulter, Brea, CA, USA). The Kolmogorov–Smirnov, D’Agostino–Pearson, and the Shapiro–Wilk tests were used to check normality of distribution. The homogeneity of variance was analyzed using the Brown–Forsythe test. Data were analyzed using Kruskal–Wallis test, followed by Mann–Whitney and Dunn’s post hoc tests for pairwise comparisons, and expressed as median and interquartile ranges [Q25; Q75]. The null hypothesis was rejected at *p* < 0.05. See the main text for significance values shown in Appendix A.

## 3. Results

### 3.1. ADI Decreased Culture Medium Arginine and Suppressed PBMC Activation and IL-2 Production

Preliminary, we studied whether and at what time point there was a change in the concentration of arginine in PBMC supernatants. After that, we assessed how the dynamics of arginine concentration correlated with cell activation, which was determined by the activity of mitochondrial dehydrogenases. This made it possible to further select time points at which other cellular functions were evaluated further (Figure 1A, Appendix A). Arginine concentration in control wells at various time points ranged from 609.1 [537.2; 664.9] µM up to 1015.2 [978.5; 1057.0] µM. After exposure to SDSCs derived from parental strain, the amino acid level fluctuated from 104.9 [63.9; 188.7] µM up to 261.8 [215.9; 320.0] µM, which was significantly lower than the values in the control. Arginine level in supernatants of cell culture incubated with mutant-strain-derived SDSCs did not differ from control and was significantly higher compared with those in supernatants of cell culture threated with parental strain SDSCs. Upon that, no significant difference was observed in supernatant arginine level without and with relevant cell stimulation (Appendix A). Arginine supplement resulted in elevated supernatant amino acid concentration from control cells and cells treated with mutant-strain-derived SDSCs (*p* < 0.01) (Appendix A). The arginine level significantly elevated up to 631.7 [570.6; 715.0] µM followed only by the second (on day 4) arginine supplementation (*p* < 0.05) in unstimulated cell culture exposed to parental-strain-derived SDSCs (Appendix A). The data obtained demonstrated that ADI activity of parental-strain-derived SDSCs resulted in a fivefold decline in arginine level as early as 24 h after exposure. Throughout the entire experiment, arginine level in cell culture treated with parental-strain-derived SDSCs did not decrease below 100 μM. ADI was likely to cause no full arginine depletion because of the dynamic equilibrium established between the activity of arginine hydrolyzing and resynthesizing enzymes. Arginine supplementation significantly restored amino acid concentration only in unstimulated cell culture at the 96 h time point, demonstrating a high rate of the arginine-hydrolyzing activity of ADI.

Since, it is well known that the activation of T cells makes the processes of oxidative phosphorylation increase [42], further experiments were conducted for investigating ADI effects on dynamic lymphocyte activation in the MTT test, allowing us to indirectly assess mitochondrial dehydrogenase activity and respiration. In the control without stimulation, no changed activity in mitochondrial respiration was observed at all time points (Figure 1B, Appendix A). Parental-strain-derived SDSCs exerted a weak but significant stimulatory effect only at the 96 h time point. Mutant-strain-derived SDSCs resulted in cell activation as early as 24 h post-exposure. In these conditions, cell mitochondrial dehydrogenase activity was significantly higher than in control at all time points. Enhanced cell activation, due to parental- and mutant-strain-derived SDSCs, should be related to effects of streptococcal superantigens in SDSCs, known to elicit lymphocyte polyclonal activation [43]. Arginine supplementation to unstimulated cell culture as early as 72 h increased the level of cell activation (*p* < 0.05) inhibited with parental-strain-derived SDSCs. Under cell stimulation with anti-CD2/CD3/CD28-beads, significantly increased (*p* < 0.001) mitochondrial respiration in control was already recorded in 24 h (Appendix A). Parental-strain-derived SDSCs suppressed anti-CD2/CD3/CD28-bead-triggered cell activation. After 48 h post-stimulation, a significant inhibitory effect of parental strain SDSCs was revealed. Arginine supplement restored mitochondrial respiration of cells retarded by parental strain SDSCs after 48 h (*p* < 0.05) (Appendix A).

The results obtained showed that when mediated by the parental strain, SDSC suppression of cell mitochondrial respiration was obviously related to ADI activity, because the mutant strain exerted no inhibitory effect. Moreover, arginine supplementation significantly restored cell activation suppressed by parental strain SDSCs.

There were no significant changes in IL-2 production after incubation of cells in the presence of the parent and mutant strains (Figure 1C, Appendix A). Stimulation PBMCs with anti-CD2/CD3/CD28-beads led to a significant increase in IL-2 production (*p* < 0.05) (Appendix A). Mutant strain SDSCs did not have any effect on the cytokine secretion, whereas parental strain SDSCs caused a twofold decrease in its production induced by anti-CD2/CD3/CD28-beads (*p* < 0.05) (Figure 1C). Arginine supplement slightly increased the IL-2 production suppressed by parental strain SDSCs, and decreased the cytokine production under standard conditions and in the presence of the mutant strain SDSCs, but the effect was not statistically significant.

Low arginine concentration in the presence of parental strain SDSCs prevented cells from increasing the activity of mitochondrial respiration. Functionally, the effect of amino acid deficiency became noticeable only 48 h after cell stimulation and was maximally expressed after 96 h of incubation. Therefore, in further experiments, changes in cellular parameters (T cell differentiation, autophagy, and proliferation) were investigated at these time points. 

### 3.2. ADI Inhibits Activation-Induced T Cell Differentiation

Amino acid deficiency was shown to affect T cell differentiation program [44]. Thus, we investigated change in percentage of naïve, CM, EM, and TEMRA subsets of CD4+ and CD8+ T cells after 48 and 96 h of culturing (Figure 2, Appendix A). It was found that after 48 h of incubation the percentage of naïve, CM, EM, and TEMRA CD8+ T lymphocytes in the control comprised 68.2 [64.6; 70.9]%, 7.8 [6.0; 13.1]%, 12.5 [10.3; 16.8]% and 9.6 [9.1; 15.8]%, respectively (Figure 2A, Appendix A). After cell incubation with parental strain SDSCs, CD8+ T cell subset composition did not differ from control cell culture. In the presence of mutant strain SDSCs, the percentage of EM CD8+ T cells significantly increased vs. control. Arginine supplementation also had no effect on the ratio of CD8+ T cell subsets in all culture settings.

After 48 h of cell stimulation with anti-CD2/CD3/CD28-tagged beads, there was a significant increase in the proportion of TEMRA to 19.6 [11.3; 26.3]% (*p* < 0.05) (Appendix A), whereas mutant strain SDSCs not only significantly increased percentage of TEMRA up to 24.5 [15.9; 30.4]% (*p* < 0.01), but also decreased proportion of naïve CD8+ T cells down to 51.1 [47.2; 71.8]% (*p* < 0.05). Cell stimulation simultaneously with parental strain SDSC treatment increased TEMRA CD8+ T cell subset up to 18.4 [16.4; 24.4] (*p* < 0.05). Arginine supplementation did not significantly affect CD8+ T cell subset composition in all culture conditions at this time point. The 96 h of culturing resulted in spontaneous alternation in T cell subset composition that featured a significantly decreased percentage of naïve CD8+ T cells down to 55.7 [41.7; 66.7]% (*p* < 0.05) and increased percentage of CM up to 11.2 [8.7; 18.5]% (*p* < 0.05) (Appendix A). Parental-strain-derived SDSCs significantly decreased CM CD8+ T cell percentage down to 8.6 [7.1; 12.8]% compared to control (*p* < 0.05). Mutant-strain-derived SDSCs exerted no effect on CD8+ T cell subset composition. Arginine supplementation did not affect the parameter studied, either (Figure 2A).

Stimulation with anti-CD2/CD3/CD28-beads for 96 h was followed by a significant decrease in percentage of naïve and TEMRA (30.3 [22.5; 51.5]% (*p* < 0.01) and 5.1 [3.8; 8.7]%, respectively (*p* < 0.05)), along with an increase in level of CM CD8+ T cells (34.5 [28.3; 46.2]% (*p* < 0.001)) compared to intact CD8+ T cells (Appendix A). The stimulated cells exposed to mutant strain SDSCs were found to undergo similar changes, and proportion of CD8+ T cell subsets did not differ from those observed in control. However, after anti-CD2/CD3/CD28-bead-based stimulation in the presence of parental strain SDSCs, percentage of CM significantly decreased (9.3 [5.8; 14.0]%), whereas the percentage of naïve (49.2 [42.3; 60.6]%) and TEMRA (21.2 [14.7; 30.3]%) CD8+ T cells, on the contrary, was elevated vs. control (Figure 2B). Upon anti-CD2/CD3/CD28-bead stimulation alongside with parental strain SDSCs, arginine supplement (Figure 2B) resulted in significantly elevated level of CM CD8+ T cells (28.3 [10.5; 39.2]% (*p* < 0.01)), whereas the level of naïve and TEMRA CD8+ T cells was found to be significantly decreased down to 37.5 [16.7; 49.1]% (*p* < 0.05) and 12.1 [4.2; 22.7]% (*p* < 0.05), respectively, compared to arginine-free settings (Appendix A). That made the T cell subset proportion closer to the control level.

After 48 h cell culturing in standard conditions, the following composition of CD4+ T cell subsets made up 56.1 [47.3; 58.2]% for naïve, 33.3 [27.4; 46.7]% for CM, 11.0 [8.9; 11.5]% for EM, and 0.7 [0.5; 1.3]% for TEMRA (Figure 3A, Appendix A). Among CD4+ T lymphocytes, the proportion of cells with the CM phenotype was significantly higher and, conversely, the proportion of cells with the TEMRA phenotype was significantly lower compared with CD8+ T cell subset profile, which is consistent with the earlier observations [45,46,47]. Mutant strain SDSCs significantly elevated naïve and decreased EM CD4+ T cell percentage (66.1 [59.5; 70.4]% and 6.5 [4.6; 9.6]%, respectively vs. control (*p* < 0.01) (Figure 3A). Parental strain SDSCs did not affect CD4+ T cell subset profile. Arginine supplement also had no effect on CD4+ T cell composition at this time point.

The 48 h long anti-CD2/CD3/CD28-bead-induced stimulation resulted in increased percentage of TEMRA CD4+ T cells up to 2.0 [1.5; 2.5]% (*p* < 0.01) (Appendix A). Under stimulation, mutant strain SDSCs did not affect the subset composition of CD4+ T cells; however, parental strain SDSCs elevated naïve (up to 68.4 [65.6; 77.1]%) and reduced EM CD4+ T cell proportion (down to 4.1 [2.8; 7.7]%) vs. control (Figure 3A). Arginine supplementation leveled the effect of parental strain SDSCs, shifting CD4+ T cell subset composition with higher percentage of EM and TEMRA (11.5 [7.9; 15.6]% (*p* < 0.001) and 4.6 [2.2; 7.3]% (*p* < 0.05), respectively) and lower percentage of naïve CD4+ T cells (52.0 [42.2; 58.0] (*p* < 0.05)).

After 96 h of incubation in standard conditions, percentage of CD4+ T cell subsets did not change considerably and was as follows: naïve—43.1 [37.1; 49.3]%, CM—45.3 [36.5; 48.9]%, EM—12.3 [9.3; 15.2]%, and TEMRA—0.8 [0.4; 1.0]% (Appendix A). Similar duration of incubation together with mutant strain SDSCs did not affect CD4+ T cell profile, whereas exposure to parental strain SDSCs elevated percentage of TEMRA CD4+ T cells up to 1.4 [0.9; 1.9]% vs. control (Figure 3B). Arginine supplement showed no remarkable effect.

After 96 h of stimulation with anti-CD2/CD3/CD28-beads, profound changes in CD4+ T cell profile were observed so that percentage of naïve cells decreased down to 16.0 [8.7; 20.4]% (*p* < 0.001), whereas percentage of CM and TEMRA increased up to 66.5 [54.2; 71.2]% (*p* < 0.001) and 1.2 [0.7; 2.0]% (*p* < 0.05), and percentage of EM remained unchanged (Appendix A). In this setting, arginine supplement caused a significant decline in level of TEMRA down to 0.6 [0.3; 1.0]% (*p* < 0.05) (Figure 3B). Parental strain SDSCs damped T cell differentiation, and the percentage of naïve cells, CM, and EM did not virtually change, while the TEMRA level significantly increased up to 3.4 [2.2; 5.9] (*p* < 0.001) compared with unstimulated cells under the same conditions (Appendix A). Upon that, arginine supplement partially canceled the inhibitory effect of the parental strain SDSCs, resulted in a decrease in the percentage of naïve and TEMRA (22.7 [13.6; 32.1]% (*p* < 0.001) and 2.0 [0.9; 3.5]% (*p* < 0.001), respectively), and elevated the level of CM cells (56.8 [48.1; 70.2]% (*p* < 0.001)) (Figure 3B). After 96 h of anti-CD2/CD3/CD28-bead stimulation in the presence of mutant strain SDSCs, a CD4+ T cell subset composition similar to the one in the control was observed.

Comparing the effects of parental and mutant strain SDSCs revealed that ADI promoted TEMRA but suppressed CM CD4+ and CD8+ T cell differentiation. The restoration of ADI-altered profile of T cell subsets to a level close to the control with arginine supplementation proved that the effect of enzyme must be related to decreased amino acid bioavailability. The ADI-mediated rise in percentage of TEMRA T cells showed that differentiation along this pathway is arginine-independent.

### 3.3. ADI Inhibited Anti-CD2/CD3/CD28-Bead Driven Proliferation of CD8+ and CD4+ T Cell Subset

Activation-induced T cell differentiation is closely related to their proliferation [44]. In fact, proliferation and activation processes in T cells are difficult to separate. In further studies, the ADI effect on different T cell subset proliferation was studied. To achieve this, we compared the division index (DI) of stimulated and unstimulated cells after incubation with parental- and mutant-strain-derived SDSCs (Figure 4).

DI of CD8+ T cell subsets without stimulation differed insignificantly, with the cells, on average, proceeding two to three divisions (Figure 4A, Appendix A). Parental and mutant strain SDSCs did not exert any effect on CD8+ T cell subset proliferation. Anti-CD2/CD3/CD28-bead stimulation significantly enhanced proliferation of naïve, CM, EM, and TEMRA CD8+ T cells, with DI comprising 9.1 [4.3; 11.5]%, 12.1 [7.7; 16.3]%, 9.7 [8.3; 15.5]%, and 8.5 [4.4; 13.4]%, respectively (*p* < 0.05 for all comparisons) (Appendix A). In this case, the CM subset underwent the highest division rate. Arginine supplement significantly reduced proliferation of naïve and TEMRA CD8+ T cells (DI 3.8 [2.9; 6.9]% and 4.7 [3.0; 6.0]%, respectively, *p* < 0.05) (Figure 4A). Anti-CD2/CD3/CD28-bead stimulation along with mutant strain SDSCs revealed no significant difference in DI for all CD8+ T cell subsets vs. control stimulation. Arginine supplement significantly (*p* < 0.05) suppressed TEMRA CD8+ T cell proliferation in this setting. Parental strain SDSCs completely suppressed anti-CD2/CD3/CD28-bead-stimulated proliferation of all CD8+ T cell subsets, with DI for naïve, CM, EM, and TEMRA lymphocytes—2.1 [2.1; 3.6]%, 1.0 [0.0; 2.4]%, 2.0 [2.0; 2.2]%, and 1.0 [0.0; 2.0]%, respectively. Arginine supplement under these conditions was followed by significantly increased proliferation of CM, EM, and TEMRA with DI of 3.8 [2.4; 11.1]% (*p* < 0.01), 4.2 [3.2; 10.1]% (*p* < 0.001), and 3.3 [2.0; 6.5]% (*p* < 0.05), respectively.

DI of CD4+ T cell subsets without stimulation made up, on average, two to four divisions, without considerable alteration after cell exposure to parental and mutant strain SDSCs, with or without arginine supplement (Figure 3B, Appendix A). Anti-CD2/CD3/CD28-bead stimulation significantly increased proliferation of CM, EM, and TEMRA CD4+ T cells, with DI increased from 3.9 [2.2; 4.1]%, 2.2 [2.0; 4.4]%, and 2.0 [0.5; 4.9]% in control—up to 8.1 [7.1; 9.7]% (*p* < 0.001), 8.9 [8.3; 10.8]% (*p* < 0.01), and 4.7 [3.9; 7.9]% (*p* < 0.05), respectively (Appendix A). Arginine supplement had no remarkable effect on CD4+ T cell proliferation upon stimulation (Figure 4B). Cell stimulation with exposure to mutant strain SDSCs was followed by the same changes in proliferative activity of CD4+ T cell subsets as in the stimulation control. Parental strain SDSCs blocked anti-CD2/CD3/CD28-bead-induced proliferation of CM, EM, and TEMRA CD4+ T cell subset (Figure 4B), so that the relevant DI was 2.1 [2.0; 4.0]%, 2.1 [2.0; 2.3]%, and 2.1 [2.0; 2.1]%. Arginine supplementation significantly restored cell division up to DI values of 5.3 [2.4; 9.0]% (*p* < 0.05), 6.5 [4.2; 9.5]% (*p* < 0.001), and 3.3 [2.7; 3.6]% (*p* < 0.001), respectively.

### 3.4. ADI Affects CD8+ and CD4+ Memory T Cell Autophagy

Autophagy is an important process regulating T cell activation and differentiation [48]. Autophagy plays a critical role in the generation of EM CD8+ T cells [49]. At the same time, the role of autophagy in CD4 T cell differentiation is not well explored. In this study, the effect of ADI on autophagy processes in T cells was investigated. Without stimulation, the level of autophagy in all CD8+ T cell subsets was approximately at equal levels comprising 3.4 [3.1; 3.8] MFI for naïve lymphocytes, 3.9 [2.6; 4.3] MFI for CM, 3.6 [2.8; 4.1] MFI for EM, and 4.0 [3.2; 5.4] MFI for TEMRA cells (Appendix A). Parental and mutant strain SDSCs as well as arginine supplement had no considerable effect on the parameter investigated (Figure 5A). Forty-eight hours after anti-CD2/CD3/CD28-bead stimulation, there was a significant increase autophagy in naïve, CM, EM, and TEMRA CD8+ T cells up to 5.8 [4.3; 6.7] (*p* < 0.01), 6.2 [5.2; 7.2] (*p* < 0.01), 6.0 [5.2; 7.4] (*p* < 0.01), and 6.1 [5.9; 7.0] (*p* < 0.05) MFI, respectively (Appendix A). When cells were stimulated with anti-CD2/CD3/CD28-bead along with mutant strain SDSCs, a control-like autophagy rise was observed, but the level of autophagy for EM CD8+ T cells was higher (6.2 [4.9; 8.1] MFI) vs. stimulation control (*p* < 0.05) (Figure 5A, Appendix A). After anti-CD2/CD3/CD28-bead stimulation in the presence of parental strain SDSCs, CM and EM CD8+ T cell autophagy level was significantly lower compared to the stimulation control, comprising 4.9 [4.3; 5.3] MFI and 4.1 [3.7; 5.4] MFI (*p* < 0.05), respectively. Administration of arginine supplement in this setting (Figure 5B, Appendix A) restored autophagy in CM and EM CD8+ T cells up to 6.1 [5.4; 8.0] MFI (*p* < 0.05) and 6.7 [6.3; 8.7] MFI (*p* < 0.01), respectively.

Ninety-six hours after anti-CD2/CD3/CD28-bead stimulation, the autophagy level reached 7.2 [6.0; 10.9], 6.9 [5.6; 11.2], 6.7 [5.1; 8.3], and 5.8 [4.9; 7.4] MFI for naïve, CM, EM, and TEMRA CD8+ T cells, respectively (*p* < 0.01 for all comparisons). This was still significantly higher compared to unstimulated cell control (Appendix A). At this time point, the intensity of autophagy for all CD8+ T cell subsets treated with anti-CD2/CD3/CD28-bead in the presence of mutant strain SDSCs did not differ from the parameters found in the stimulation control and was significantly higher compared to the autophagy level in unstimulated cells treated with mutant strain SDSCs (Appendix A). Ninety-six hours after stimulation, the autophagy-inhibiting effect of parental strain SDSCs diminished, so that the differences between the autophagy level in the stimulation control and stimulation along with parental strain SDSCs became statistically insignificant. Arginine supplement had no significant effect on autophagy at this time point.

In control, after 48 h of incubation, all CD4+ T cell subsets had approximately similar autophagy levels, comprising 2.9 [2.3; 3.4] MFI for naïve cells, 3.2 [2.8; 3.7] MFI for CM cells, 3.2 [2.7; 4.2] MFI for EM, and 4.1 [3.3; 4.9] MFI for TEMRA (Appendix A). Cells exposed to parental and mutant strain SDSCs as well as arginine supplement did not significantly alter intensity of this process. Forty-eight hours after anti-CD2/CD3/CD28-bead stimulation, there was a significant increase in the level of autophagy up to 5.7 [4.2; 6.3] MFI in naïve lymphocytes, 5.9 [4.8; 7.0] MFI in CM cells, 6.1 [5.9; 9.6] MFI in EM cells (*p* < 0.01 for all comparisons), and 7.9 [6.7; 9.5] MFI (*p* < 0.05) in TEMRA (Appendix A). After cell stimulation and mutant strain SDSC treatment, similar changes were found, and the level of autophagy did not differ from the corresponding magnitude in the stimulation control (Figure 5B). Upon anti-CD2/CD3/CD28-bead stimulation along with parental strain SDSC treatment, the autophagy level of CM, EM, and TEMRA CD4+ T cells was significantly lower than the stimulation control, comprising 4.0 [3.6; 4.8] MFI, 4.8 [3.6; 5.2], and 4.9 [3.4; 5.9] MFI, respectively (*p* < 0.05 for all comparisons). Arginine supplement was followed by restoration of autophagy, inhibited by parental strain SDSCs, up to 5.6 [5.5; 7.3] (*p* < 0.01) for CM, 7.4 [6.8; 10.4] (*p* < 0.01) for EM, and 8.2 [7.9; 9.8] (*p* < 0.05) for TEMRA CD4+ T cells.

Ninety-six hours after anti-CD2/CD3/CD28-bead stimulation, the level of autophagy in CD4+ T cells was significantly higher compared to that in the unstimulated cells and reached 5.6 [4.3; 6.9] in naïve lymphocytes, 6.3 [5.5; 10.2] in CM cells, 6.7 [5.6; 9.0] in EM cells (*p* < 0.01 for all comparisons), and 7.2 [5.8; 8.1] (*p* < 0.05) in TEMRA cells (Appendix A). Stimulated cell treatment with parental and mutant strain SDSCs had no significant effect on the autophagy level vs. corresponding values in the stimulation control. Arginine supplement also did not have a significant effect on tested parameters at this time point.

Thus, ADI inhibited autophagy in CM and EM CD8+ T cells and CM, EM, and TEMRA CD4+ T cell subsets in 48 h after anti-CD2/CD3/CD28-bead-based stimulation. The effect of the enzyme was due to the arginine depletion in the culture medium.

## 4. Discussion

One of the most evolutionarily ancient strategies used by microbes to interfere with immune response and immune evasion is based on nutrient depletion in the microenvironment of host cells [21,50]. Here, we investigated an effect of bacterial arginine-hydrolyzing enzyme, ADI, on human peripheral blood T cell activation and differentiation. Comparing the impact of parental and mutant strain, SDSCs showed that ADI suppresses anti-CD2/CD3/CD28-bead-induced activation (Figure 1B and Appendix A) and affects T memory cell subset composition (Figure 2 and Figure 3 and Appendix A). After 96 h of activation, parental strain SDSCs markedly damped anti-CD2/CD3/CD28-bead induced CM CD8+ T cell and CM, EM CD4+ T cell differentiation, but elevated TEMRA CD8+ and CD4+ T cell, so cell subset ratio did not differ profoundly vs. unstimulated cells (Figure 2 and Figure 3 and Appendix A). Proliferation magnitude for all T cell subsets treated with parental strain SDSCs remained at a level observed for intact T cells (Figure 4, Appendix A). Mutant strain SDSCs had no considerable effect on CD8+ and CD4+ T cell subset composition and proliferation.

The results confirm the importance of arginine for CM differentiation and are in line with findings obtained by Geiger et al. [20]. In his work, it was shown that 24 and 48 h after T cell activation, even despite the high concentration of arginine in the culture medium (1 mM) and high rate of arginine uptake by activated CD8+ T cells, there was a sharp decrease in intracellular arginine level. The intracellular concentration of all other amino acids either remained stable or increased. A rise in intracellular arginine level shifted human and mouse T cells towards the CM-like phenotype, characteristic of high CCR7 and CD62L expression, and lowered IFN-γ production as well as retarded glycolysis [20]. Studies of the role of arginine in CD4+ T cell differentiation have not yet been conducted. Obviously, the effect of parental strain SDSCs was due to ADI’s capacity to deplete arginine in T cell microenvironments because arginine supplement decreased percentage of naïve and TEMRA CD4+ and CD8+ T cells, paralleled with a rise in CM cells (Figure 2 and Figure 3 and Appendix A).

Amino acid deficiency in T cells blocks protein synthesis owing to accumulated uncharged tRNAs, which leads to activation of general control nonderepressible 2 (GCN2) stress kinase, followed by phosphorylation of the eukaryotic translation initiation factor 2A (eIF2A) and reduced translation rate. Functionally, GCN2 activation results in a proliferation arrest, induces anergy, and impairs CD8+ T cell cytotoxic effector functions in mice [3].

mTOR is another arginine-dependent intracellular cascade [51,52] that integrates T cell activation signals and controls long-term consequences of antigen recognition [53]. mTOR in T cells activates several downstream effector pathways, important for productive immune response, including immune receptor signaling, metabolic programs, and migratory activity [54]. In T cells, mTOR was identified as one of the downstream targets of the IL-2 signaling pathway, which regulates Cdk2 and Cdc2 kinases as well as cell cycle progression. Our studies confirm a decrease in IL-2 production under arginine deficiency (Figure 1C, Appendix A), which should be one of the mechanisms for suppressing the activation of T cells.

Metabolic profile is crucial for T memory cells formation [55,56,57] and is also under mTOR-axis control [58,59]. Using siRNA for mTOR, Raptor or FKBP12 inhibition revealed mTORC1 involvement in antigen-specific CD8+ T cell differentiation [60]. mTOR is suggested to be critical for regulating CD4+ T cell antigen sensitivity [53]. mTOR inhibition with rapamycin was shown to result in clonal anergy of activated Th1 cells [53]. ADI-mediated arginine depletion in culture medium upon T cell activation can be responsible for inhibition of arginine-dependent mTOR signaling, disruption of metabolic switch from oxidative phosphorylation to glycolysis, and downregulation of IL-2 proliferation signal.

Arginine is the primary amino acid that regulates mTORC1 in several cell types, including human embryonic stem cells (hESCs). Dependence on arginine persists after hESCs differentiate into fibroblasts, neurons, and hepatocytes, highlighting the fundamental importance of arginine for mTORC1 signaling [51]. The effect of arginine deficiency on the activation of mTOR signaling cascade in T cells has not been studied well enough. In a single work, devoted to the molecular mechanisms of T cell activation under arginine starvation, it was shown that the lymphocyte culturing in arginine-depleted medium suppressed the Akt/mTOR signaling pathway and induced ER-stress-mediated autophagy [61]. Autophagy inhibition under arginine deficiency led to cell death, whereas amino acid replenishment restored normal cell cycle profile and proliferation [61]. According to the generally accepted concept, during cell starvation, mTOR and autophagy are reciprocally regulated. Particularly, nutrient-deficiency-induced inhibition of the mTORC1 leads to increased autophagy [62]. However, in T cells, activation-induced autophagy is upregulated in an mTOR-independent manner [49,63]. Inhibition of the mTOR signaling cascade with rapamycin has a minor effect on macroautophagy in activated T cells [49]. On the contrary, autophagy replenishes elevated cell demand in nutrients and supports mTORC1 activation [64]. During T cell differentiation, autophagy removes accumulated nonfunctional mitochondria and damaged proteins to decline oxidative stress and improve viability in effector cells undergoing multiple divisions [37,49,65,66]. Autophagy is also involved in metabolism regulation [37] and T cell tolerance induction [67,68], and it modulates TCR-induced NF-kB activation in effector T cells [37,69]. Chaperone-mediated autophagy induced in activated T cells was shown to control TCR signaling [37].

Our study revealed ADI-mediated suppression autophagy in activated T cells. Particularly, autophagy depression was observed in CM and EM of CD8+ T cell subsets (Figure 5A, Appendix A), as well as CM, EM, and TEMRA of CD4+ subsets (Figure 5B, Appendix A) in 48 h after stimulation with anti-CD2/CD3/CD28-beads. Strikingly, the opposite results were obtained in the study by García-Navas et al. on the molecular mechanisms of autophagy regulation during the activation of T lymphocytes in conditions of arginine deficiency [61]. Using genetic and biochemical approaches, it was found that the lymphocyte cultivation in an arginine-deficient medium led to the suppression of the Akt/mTOR signaling pathway, the development of ER stress, and increased autophagy. Inhibition of autophagy in arginine-free conditions caused cell death, and replenishment of the amino acid restored cell cycling and proliferation [61].

Why T cell activation in an arginine-free environment induces autophagy, whereas enzymatic arginine depletion blocks autophagy, requires further investigation. Perhaps this is due to the fact that ADI does not cause absolute deficiency of arginine in the cell culture medium (Figure 1A and Appendix A). It was shown that the amino acid concentration in cell culture supernatants in the presence of the enzyme did not fall below 100 µM. It is quite likely that such a decrease in arginine level does not result in cell starvation, but can disrupt arginine-dependent processes: dynamics of actin cytoskeleton, mTOR-mediated TCR signaling, and signal transduction downstream of co-stimulatory molecules or cytokine receptors. Recent studies have shown that cytokines including IL-2, IL-4, IL-7, and IL-15, signaling via the common γ-chain, activate and maintain a high level of macroautophagy in a JAK-mediated manner in CD4+ T cells for sustained proliferation and survival as well as effector and memory cell differentiation [49,70]. Our studies demonstrated that arginine supplementation was beneficial and ameliorated the functions of T cells disrupted by ADI, but the impaired cellular functions failed to restore up to the initial values. This is quite comprehensible, because the enzyme constantly hydrolyzed the amino acid, which studies of the dynamics of arginine concentration in cell culture supernatants clearly demonstrated (Figure 1A, Appendix A). In the presence of the parental strain SDSCs, the arginine supplement leads to a slight increase in the level of arginine, but still, the concentration of amino acid in the culture medium does not reach the control level.

T cell activation is accompanied by large-scale rearrangements of the actin cytoskeleton playing a decisive role in immune synapse formation and, hence, lymphocyte activation [71,72,73]. Arginine deprivation results in impaired cofilin-mediated actin dynamic regulation and decreased CD2 and CD3 accumulation in the developing immune synapse [74]. These data suggest that ADI-mediated inhibition of T cell differentiation may be related to defective immune synapse formation, weak TCR and costimulatory signals, and reduced proliferation and cytokine production in arginine-deficiency settings.

## 5. Conclusions

The studies conducted show that streptococcal ADI suppresses CM CD4+ and CD8+ T cell activation and differentiation. The effect of ADI on T cells may be associated with the disruption of multiple cellular processes depending on arginine bioavailability, including intracellular signaling cascades, autophagy, and cytoskeletal rearrangement. All the processes are not only dependent on arginine availability but also closely related and affect each other. This complicates the understanding of the mechanisms underlying the enzyme’s action on T cells. The fact that streptococcal ADI suppresses the differentiation of memory cells can explain the ability of this pathogen to cause chronic recurrent infections, as well as persistent streptococci in tonsils. Further investigation of bacterial ADI is helpful to expand the existing understanding of the mechanisms of pathogen immune evasion. ADI biopotential in tumor regression is well known. Pegylated recombinant mycoplasma ADI (ADI-PEG 20) is in the last phase of clinical trials against various arginine auxotrophic cancers, such as hepatocellular carcinoma, melanoma, and mesothelioma [75]. Recently, ADI has become of great importance in the treatment of Alzheimer’s disease and antiviral drugs [76]. Arginine catabolism with ADI-PEG 20 is considered a promising method of obesity treatment and related disorders [77]. It is necessary to take into account that ADI-based therapy can be accompanied by serious complications due to suppression of immunity. However, the approach should be a success for reprogramming adaptive immune responses to look after conditions with unwanted immune system hyperreactivity.

## Figures and Tables

**Figure 1 microorganisms-11-02585-f001:**
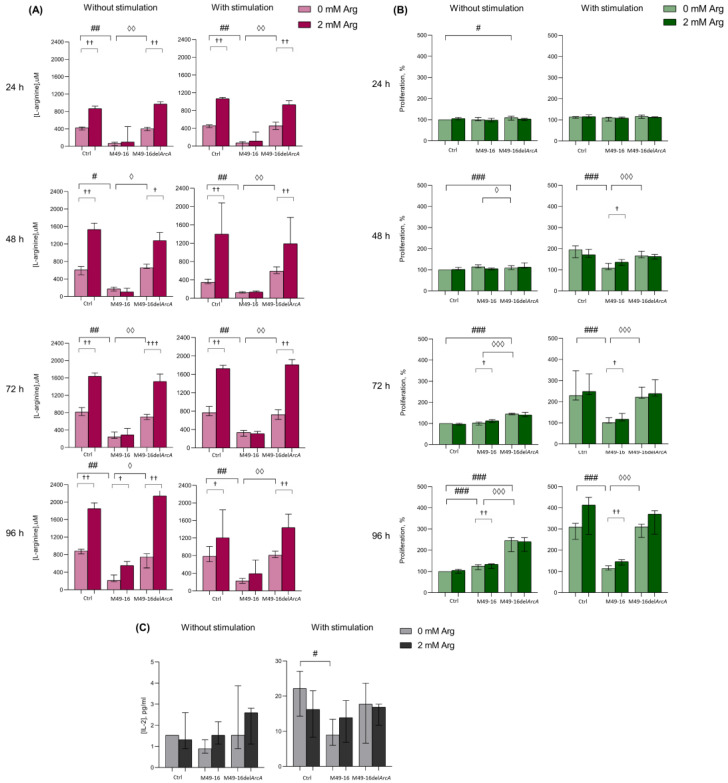
Effects of parental- and mutant-strain-derived SDSCs on PBMCs. (**A**) Time-dependent arginine concentration in cell culture supernatants after exposure to parental- and mutant-strain-derived SDSCs; (**B**) Dynamics of PBMC activation after exposure to parental- and mutant-strain-derived SDSCs; (**C**) IL-2 concentration in cell culture supernatants after exposure to parental- and mutant-strain-derived SDSCs. PBMCs were stimulated with anti-CD2/CD3/CD28-bead in the presence of parental- or mutant-derived SDSCs, with or without arginine supplement. PBMC activation was assessed in MTT assay. The assessment of IL-2 concentration was performed using a human IL-2 Enzyme-Linked Immunosorbent Assay. Data were analyzed using Kruskal–Wallis test (*p* < 0.001), followed by Mann–Whitney test for pairwise comparisons, and expressed as median and interquartile ranges [Q25; Q75], (for (**A**,**B**) n = 8, for (**C**) n = 7). The differences are significant: vs. control, ### *p* < 0.001, ## *p* < 0.01, # *p* < 0.05; vs. SDSC M49-16, ◊◊◊ *p* < 0.001, ◊◊ *p* < 0.01, ◊ *p* < 0.05; vs. arginine supplement, ††† *p* < 0.001, †† *p* < 0.01, † *p* < 0.05.

**Figure 2 microorganisms-11-02585-f002:**
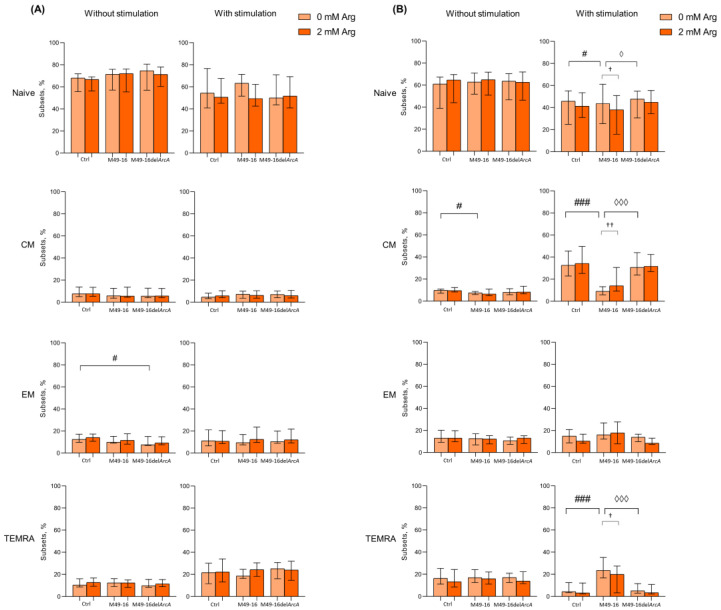
Effects of parental- and mutant-strain-derived SDSCs on CD8+ T cell differentiation. (**A**) Proportion of CD8+ T cell subsets after 48 h of incubation; (**B**) Proportion of CD8+ T cell subsets after 96 h of incubation. PBMCs were stimulated with anti-CD2/CD3/CD28-bead in the presence of parental- or mutant-derived SDSCs, with or without arginine supplement. Cell suspensions were stained with antibodies cocktail. Samples were analyzed flow cytometry. DAPI-negative cells were used for analysis. Naïve cells were defined as CD45RA+CD62L+, central memory cells (CM) as CD45RA−CD62L+, effector memory cells (EM) as CD45RA−CD62L− and terminally differentiated effector memory cells (TEMRA) as CD45RA+CD62L−. Data were analyzed using Kruskal–Wallis test (*p* < 0.001), followed by Mann–Whitney test for pairwise comparisons and expressed as median and interquartile ranges [Q25; Q75] (n = 8). The differences are significant: vs. control, ### *p* < 0.001, # *p* < 0.05; vs. SDSC M49-16, ◊◊◊ *p* < 0.001, ◊ *p* < 0.05; vs. arginine supplement*,* †† *p* < 0.01, † *p* < 0.05.

**Figure 3 microorganisms-11-02585-f003:**
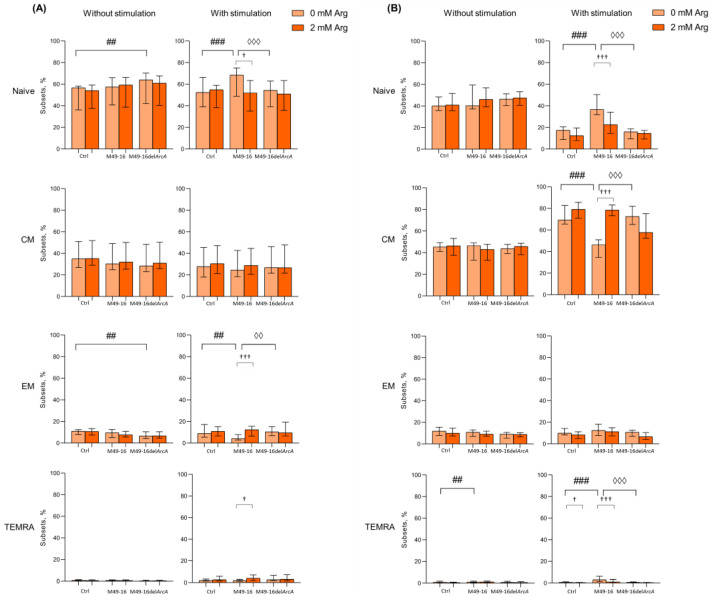
Effects of parental- and mutant-strain-derived SDSCs on CD4+ T cell differentiation. (**A**) Proportion of CD4+ T cell subsets after 48 h of incubation; (**B**) Proportion of CD4+ T cell subsets after 96 h of incubation. PBMCs were stimulated with anti-CD2/CD3/CD28-bead in the presence of parental- or mutant-derived SDSCs, with or without arginine supplement. Cell suspensions were stained with antibodies cocktail. Samples were analyzed by flow cytometry. DAPI-negative cells were used for analysis. Naïve cells were defined as CD45RA+CD62L+, central memory cells (CM) as CD45RA−CD62L+, effector memory cells (EM) as CD45RA−CD62L−, and terminally differentiated effector memory cells (TEMRA) as CD45RA+CD62L−. Data were analyzed using Kruskal–Wallis test (*p* < 0.001), followed by Mann–Whitney test for pairwise comparisons, and expressed as median and interquartile ranges [Q25; Q75] (n = 8). The differences are significant: vs. control, ### *p* < 0.001, ## *p* < 0.01,; vs. SDSC M49-16, ◊◊◊ *p* < 0.001, ◊◊ *p* < 0.01; vs. arginine supplement, ††† *p* < 0.001, † *p* < 0.05.

**Figure 4 microorganisms-11-02585-f004:**
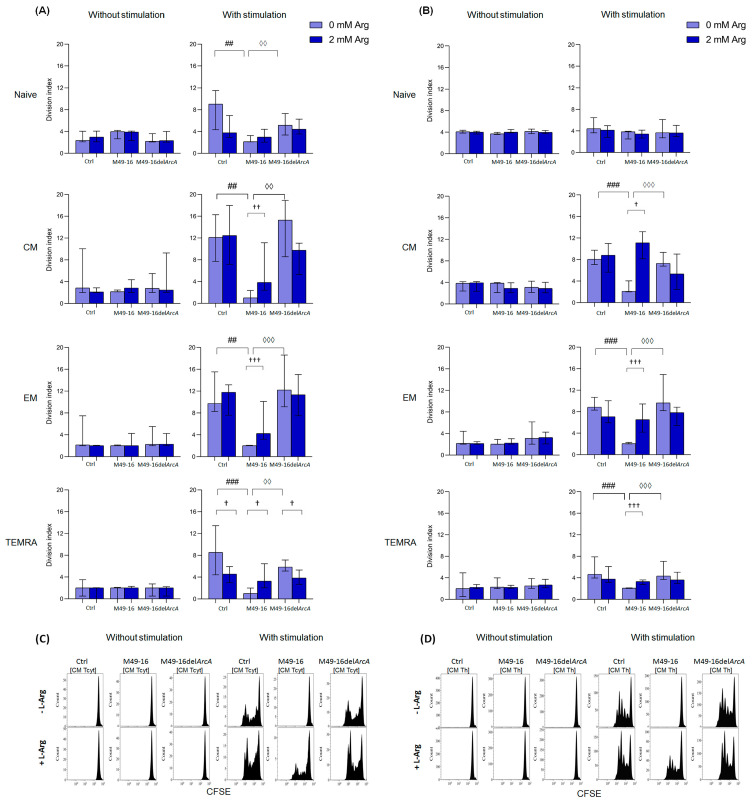
Effects of parental- and mutant-strain-derived SDSCs on T cell subsets proliferation. (**A**) CD8+ T cell proliferation; (**B**) CD4+ T cell proliferation; (**C**) Representative histograms reflecting changes in the CM CD8+ T cell proliferation; (**D**) Representative histograms reflecting changes in the CM CD4+ T cell proliferation. CFSE-labeled PBMCs were stimulated with anti-CD2/CD3/CD28-bead in the presence of parental- or mutant-derived SDSCs, with or without arginine supplement. T cell proliferation was determined by flow cytometry with Navios software 2.0. FCS Express 7 (De Novo Software) demo software was used to analyze flow cytometry data, expressed as division index (DI). Data were analyzed using Kruskal–Wallis test (*p* < 0.001), followed by Mann–Whitney test for pairwise comparisons, and expressed as median and interquartile ranges [Q25; Q75] (n = 8). The differences are significant: vs. control, ### *p* < 0.001, ## *p* < 0.01; vs. SDSC M49-16, ◊◊◊ *p* < 0.001, ◊◊ *p* < 0.01; vs. arginine supplement, ††† *p* < 0.001, †† *p* < 0.01, † *p* < 0.05.

**Figure 5 microorganisms-11-02585-f005:**
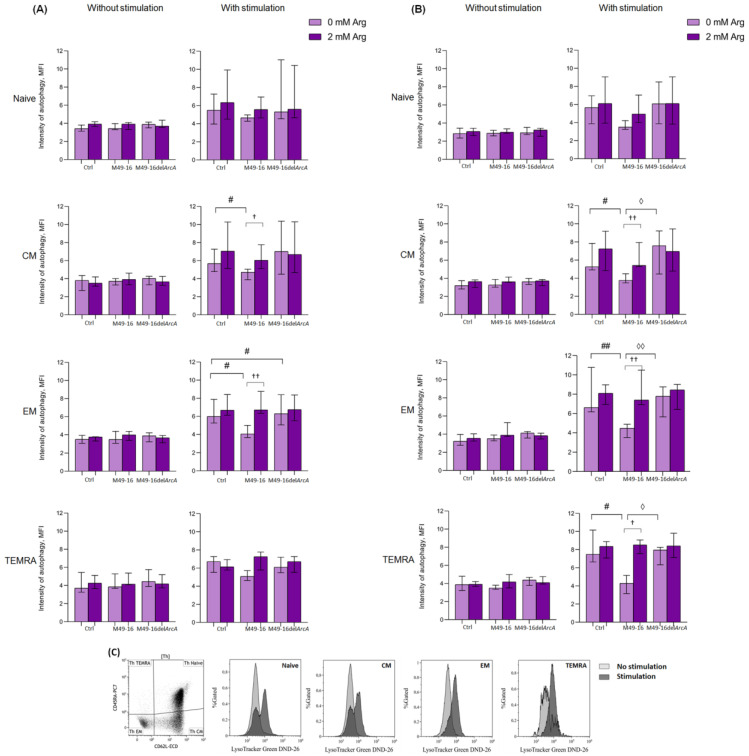
Effects of parental- and mutant-strain-derived SDSCs on T cell autophagy. (**A**) CD4+ T cell autophagy after 48 h of incubation; (**B**) CD4+ T cell autophagy after 48 h of incubation; (**C**) Representative histograms reflecting changes in the autophagy level in the control during anti-CD2/CD3/CD28-induced stimulation of CD4+ T cell. PBMCs were stimulated with anti-CD2/CD3/CD28-bead in the presence of parental- or mutant-derived SDSCs, with or without arginine supplement. To determine T memory subset autophagy level, cell suspensions were stained with an antibodies cocktail, as described above, and Lysotracker Green. To exclude necrotic cells from the analysis, the samples were stained with DAPI. Samples were analyzed by flow cytometry. The results were expressed as mean of fluorescence intensity (MFI). Data were analyzed using Kruskal–Wallis test (*p* < 0.001), followed by Mann–Whitney test for pairwise comparisons, and expressed as median and interquartile ranges [Q25; Q75] (n = 8). The differences are significant: vs. control, ## *p* < 0.01, # *p* < 0.05; vs. SDSC M49-16, ◊◊ *p* < 0.01, ◊ *p* < 0.05; vs. arginine supplement, †† *p* < 0.01, † *p* < 0.05.

## Data Availability

The data presented in this study are available on request from the corresponding author.

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
