# Peer review of "Streptococcal Arginine Deiminase Inhibits T Lymphocyte Differentiation In Vitro"

_microorganisms, 2023, doi:10.3390/microorganisms11102585_

Round 1
Reviewer 1 Report (Previous Reviewer 2)
Authors addressed all my comments in the previous revision.
Author Response
We sincerely thank Reviewer #1 for the positive assessment of our work!
Reviewer 2 Report (New Reviewer)
The paper “Streptococcal arginine deiminase inhibits T lymphocyte dif-ferentiation in vitro” by Eleonora A. Starikova et al. demonstrate the effected of arginine deiminase (ADI) was inhibition of autophagy processes in CD8+ CM and EM as well as CD4+ CM, EM and TEMRA T cell subsets. The author found arginine crucial role in controlling immune reactions. The result suggested that streptococcal ADI may downregulate adaptive immunity and immunological memory. The research is interesting and novel. The manuscript can be accepted after the following points are addressed:
1. A graphical abstract is necessary here.
2. Did authors check the PBMCs purification quality from peripheral blood samples; specify the marker, if used?
3. In figure 1, the reason of authors choose the different time points? Is it time dependent? More discussion is needed here.
4. What are the clinical applications of ADI? The authors may add appropriate discussion.
The English language should be thoroughly edited and polished.
Author Response
We sincerely thank Reviewer #2 for the positive assessment of our work, and comments that make it better!
- A graphical abstract is necessary here.
The answer: Agree, thanks for the comment!
- Did authors check the PBMCs purification quality from peripheral blood samples; specify the marker, if used?
The answer: We have added the missing information to the text (lines 130-132) and supplementary material (Figure 1S, page 1).
- In figure 1, the reason of authors choose the different time points? Is it time dependent? More discussion is needed here.
The answer: We have added the comments to the text (lines 248-253; 323-328).
- What are the clinical applications of ADI? The authors may add appropriate discussion.
The answer: We have added the discussion to the text (lines 711-716).
Comments on the Quality of English Language
The English language should be thoroughly edited and polished.
The answer: Agree, thanks for the comment! The text has been revised.
Round 2
Reviewer 2 Report (New Reviewer)
The revised version of the manuscript now seems improved. But a few points still need consolidated pieces of evidence and clarification.
This manuscript is a resubmission of an earlier submission. The following is a list of the peer review reports and author responses from that submission.
Round 1
Reviewer 1 Report
This reviewer appreciates the improvement of the revised manuscript. However, the major concerns still exist such as physiologicalirrelevance of experimental settings and indistinguishable results between T cells supplemented with arginine and T cells cultured without extra arginine (Figure 2A, C, D; Figure 3A, B; Figure 4A, C, D).
There are some redundancy of wording, could be improved with shortened and more concise sentences.
Author Response
Comments and Suggestions for Authors
This reviewer appreciates the improvement of the revised manuscript. However, the major concerns still exist such as physiological irrelevance of experimental settings and indistinguishable results between T cells supplemented with arginine and T cells cultured without extra arginine (Figure 2A, C, D; Figure 3A, B; Figure 4A, C, D).
We thank the Reviewer 1 for his attentive, critical attitude towards our work and positive assessment of the revision of the article. We also thank the Reviewer 1 for finding our study interesting and for valuable comments.
We do not state that the conditions of the experiment are physiological. We make a very accurate conclusion that bacterial arginine deiminase suppresses the differentiation of memory T cells in vitro and that arginine is important for these processes. We only assume that the enzyme can have a similar effect in vivo. And this may be a subject to further investigation.
Please note that in the latest version of the article, according to the recommendations of the other Reviewers, changes were made in the format of data presentation. Figures with numbers referred to by the Reviewer 1 (Figure 2A, C, D; Figure 3A, B; Figure 4A, C, D) are not available in the second version of the article.
The study of the effect of arginine was not the purpose of our study, but served as internal controls. Here we prove that supernatant of destroyed streptococcal cells of parental strain containing arginine deiminase suppresses T cell differentiation due to a decrease in arginine in the medium.
The absence of the effect of arginine supplementation in almost all cases shows that there is no the amino acid deficiency in the culture, and a further increase in the concentration of arginine makes no difference. The effect of arginine supplementation becomes noticeable only against the inhibitory effect of supernatant of destroyed streptococcal cells of parental strain. Arginine supplementation under these conditions restores cellular functions, as the Reviewer 1 correctly noted (Fig. 1B increases metabolism, Fig. 2B and Fig. 3B enhances differentiation of memory cells, Fig. 4A, B increases proliferation, Fig. 5A, B increases autophagy).
Comments on the Quality of English Language
There are some redundancy of wording, could be improved with shortened and more concise sentences.
As it was suggested by Reviewer 1, we carefully checked the text, the tables and figures for grammatical errors. We have really tried to make the text more laconic.
Reviewer 2 Report
In this paper the authors showed the importance of arginine in immune response and how streptococcal impact the level of this amino acid after an infection.
The study is well designed and I appreciated the presentation of all figures with heatmaps but I have some comments
It's not clear what the authors wanted to prove in Fig. 1b. In y opinion MTT assay is not a good assay for measuring cell proliferation as it measured indirectly the proliferation. Authors should use another test to prove the effect on cell proliferation.
For the autophagy part, authors must add a positive (inducer of autophagy) and negative (inhibitor of autophagy) control to be sure that their technique worked. Moreover, authors should add measurement of LC3 by western blot ton confirm these results.
Author Response
Comments and Suggestions for Authors
In this paper the authors showed the importance of arginine in immune response and how streptococcal impact the level of this amino acid after an infection.
The study is well designed and I appreciated the presentation of all figures with heatmaps but I have some comments.
It's not clear what the authors wanted to prove in Fig. 1b. In y opinion MTT assay is not a good assay for measuring cell proliferation as it measured indirectly the proliferation. Authors should use another test to prove the effect on cell proliferation.
We thank the Reviewer 2 for the friendly analysis of the work and the comments made. We also thank the Reviewer 2 for finding our study interesting and for valuable comments.
In the MTT test, the effect of arginine deiminase on peripheral blood mononuclear cells activation was assessed by the dynamics of mitochondrial dehydrogenase activity (Fig. 1B) (Mølgaard et al., 2021). The dependence of this parameter on the concentration of arginine in the culture medium was compared in order to choose the optimal time range for other tests. According to this experiment, it is clear that low concentrations of arginine correspond to a weak response of cells to activation with anti-CD2/CD3/CD28 loaded beads, starting from 72 hours (See Fig. 1 A, B). To assess proliferative activity, a method based on cell staining with a vital CFSE dye followed by an analysis of the division index was used (Fig. 4).
- Mølgaard, K.; Rahbech, A.; Met, Ö.; Svane, I. M.; Thor Straten, P.; Desler, C.; Peeters, M. J. W. Real-Time Monitoring of Mitochondrial Respiration in Cytokine-Differentiated Human Primary T Cells. J Vis Exp 2021, No. 176. https://doi.org/10.3791/62984.).
For the autophagy part, authors must add a positive (inducer of autophagy) and negative (inhibitor of autophagy) control to be sure that their technique worked.
We thank Reviewer 2 for this important suggestion. According to the literature (Li et al., 2006; Hubbard et al., 2010; Arnold et al., 2014; Botbol et al., 2015), activation of lymphocytes through TCR is always followed by increased autophagy, therefore, we considered the activation of cells with anti-CD2/CD3/CD28 loaded beads as a positive control of the induction of autophagy. Negative control was appended to the results (page 10 of Supplementary Materials, Figure 1S). We hope that the modified text and figure will make it more reader-friendly.
- Li C, Capan E, Zhao Y, Zhao J, Stolz D, Watkins SC, et al. Autophagy is induced in CD4+ T cells and important for the growth factor-withdrawal cell death. J Immunol (2006) 177:5163–8. 10.4049/jimmunol.177.8.5163.
- Hubbard VM, Valdor R, Patel B, Singh R, Cuervo AM, Macian F. Macroautophagy regulates energy metabolism during effector T cell activation. J Immunol. 2010;185:7349–7357.
- Arnold CR, Pritz T, Brunner S, Knabb C, Salvenmoser W, Holzwarth B, Thedieck K, et al. T cell receptor-mediated activation is a potent inducer of macroautophagy in human CD8(+)CD28(+) T cells but not in CD8(+)CD28(−) T cells. Exp Gerontol. 2014;54:75–83.
- Botbol Y, Patel B, Macian F. Common gamma-chain cytokine signaling is required for macroautophagy induction during CD4 T cell activation. Autophagy. 2015:0.
Moreover, authors should add measurement of LC3 by western blot ton confirm these results.
We understand the Reviewer's concern regarding autophagy analysis. Confirmation of the results obtained by Western blot would significantly have strengthened our data. In our work, we relied on studies that confirm the effectiveness of using LysoTracker to evaluate cellular autophagy. Previously (Chikte et al., 2014) it was shown that LC3 and LysoTracker measure different biological events, but both are activated during autophagy. And, in another study Alonzi et al., 2019, a good correlation of the data obtained using WB and flow cytometry was demonstrated.
- Alonzi T, Petruccioli E, Vanini V, Fimia GM, Goletti D. Optimization of the autophagy measurement in a human cell line and primary cells by flow cytometry. Eur J Histochem. 2019 Jun 26;63(2):3044. doi: 10.4081/ejh.2019.3044. PMID: 31243942; PMCID: PMC6610717.
- Chikte S, Panchal N, Warnes G. Use of LysoTracker dyes: a flow cytometric study of autophagy. Cytometry A. 2014 Feb;85(2):169-78. doi: 10.1002/cyto.a.22312. Epub 2013 Jul 11. PMID: 23847175.
We hope that the current and quite voluminous study allows us to give a preliminary assessment of the changes that occur in lymphocytes under the influence of streptococcal arginine deiminase. In further studies to be carried out, this issue will be investigated in more detail.
Round 2
Reviewer 2 Report
Authors addressed all my comments